# Antimicrobial Resistance and Prevalence of Extended Spectrum β-Lactamase-Producing *Escherichia coli* from Dogs and Cats in Northeastern China from 2012 to 2021

**DOI:** 10.3390/antibiotics11111506

**Published:** 2022-10-28

**Authors:** Yifan Zhou, Xue Ji, Bing Liang, Bowen Jiang, Yan Li, Tingyv Yuan, Lingwei Zhu, Jun Liu, Xuejun Guo, Yang Sun

**Affiliations:** 1Engineering Research Center of Glycoconjugates, Ministry of Education, School of Life Sciences, Northeast Normal University, Changchun 130024, China; 2Changchun Veterinary Research Institute, Chinese Academy of Agricultural Sciences, Changchun 130021, China; 3Key Laboratory of Jilin Province for Zoonosis Prevention and Control, Changchun 130021, China

**Keywords:** AMR, cats, dogs, *E. coli*, ESBL, MDR, phylogenetic groups

## Abstract

(1) Background: there has been a growing concern about pet-spread bacterial zoonosis in recent years. This study aimed to investigate the trend in drug-resistance of canine *Escherichia coli* isolates in northeast China between 2012–2021 and the differences in drug-resistance of *E. coli* of different origins in 2021. (2) Methods: *E. coli* were isolated from feces or anal swab samples from dogs and cats, and their antibiotic susceptibility profiles and phylogenetic grouping were identified. PCR was applied on the extended spectrum β-lactamase (ESBL) *E. coli* for antibiotic resistance genes. (3) Results: five hundred and fifty-four *E. coli* isolates were detected in 869 samples (63.75%). The multidrug resistance (MDR) rates of *E. coli* in pet dogs showed a decreasing trend, but working dogs showed the opposite trend. Resistance genes *bla*_CTX-M_ and *bla*_CTX-M+TEM_ were dominant among the ESBL producers (*n* = 219). The consistency between the resistance phenotypes and genes was high except for fluoroquinolone-resistant ESBL *E. coli*. All ESBL *E. coli*-carrying *bla*_NDM_ were isolated from working dogs, and one of the strains carried *mcr-1* and *bla*_NDM-4_. Phylogroup B2 was the dominant group in pet cats, and more than half of the isolates from companion cats were ESBL *E. coli*. (4) Conclusions: the measures taken to reduce resistance in China were beginning to bear fruit. Companion cats may be more susceptible to colonization by ESBL *E. coli*. The problem of resistant bacteria in working dogs and pet cats warrants concern.

## 1. Introduction

As the first choice for the treatment of bacterial infection, antibiotics play an important role in the clinical treatment of humans and animals [1] and can also promote the growth of livestock and poultry. Before China’s Ministry of Agriculture and Rural Development issued a ban on the addition of antibiotics to feed, nearly half of the antibiotics used every year were mixed in animal feed to improve the growth of livestock and poultry. Antimicrobial resistance has become one of the most challenging problems in public health due to the overuse of antibiotics [2].

Most *Escherichia coli* are considered important components of the intestinal flora, but some *E. coli* carrying virulence genes can act as conditional pathogens to cause intestinal or extraintestinal infections, such as diarrhea, urinary tract infections, sepsis and meningitis, and a proportion of extraintestinal pathogenic *E. coli* can inhabit the intestine as normal flora and cause extraintestinal infections [3,4]. *E. coli* is one of the most widespread organisms in the intestinal flora and is chronically exposed to antibiotic selection pressure; therefore, testing *E. coli* isolated from fecal samples for drug resistance is a good indicator of the level of resistance in the intestinal flora [5].

*Escherichia coli* is sensitive to antibiotic pressure and has a very high potential for the development of drug resistance, leading to an increasing number of multidrug-resistant (MDR) and even pandrug-resistant strains. MDR is defined as non-susceptible to at least one agent in three or more antimicrobial categories [6]. MDR pathogens can cause more severe, longer-lasting infections and ultimately have global pandemic potential [7]. *E. coli* resistance to β-lactams is usually associated with plasmid-mediated extended spectrum β-lactamase (ESBL) [8]. As a horizontal transmission vector of drug resistance genes, plasmids can be transmitted between different species of bacteria with *bla*_ESBL_. More and more *bla*_ESBL_ genes have been reported in bacteria, including commensal bacteria that cause canine diseases.

Current research related to antibiotic resistance is mainly focused on food-origin animals, such as livestock and poultry. However, because of close contact between companion animals and human beings, *E. coli* isolated from companion dogs and cats is highly similar to human isolates, increasing the possibility of passing resistant *E. coli* between species and bringing a potential threat to human health [9]. Working dogs around the world play an irreplaceable role in many tasks, including product detection, protection, search and rescue, owing to their keen sense of smell and long-lasting physical abilities [10]. However, few studies of the prevalence of drug resistance have focused on working dogs in China.

This study explored the impact of changes in antibiotic use trends in recent years on *E. coli* drug resistance phenotypes by investigating the drug resistance of *E. coli* in dogs in northeast China from 2012 to 2021, and also investigated the prevalence of drug resistance phenotypes and drug resistance genes of ESBL *E. coli* in dogs and cats, as well as enriching the drug resistance data related to *E. coli* in working dogs.

## 2. Results

### 2.1. E. coli Isolation

The isolates of *E. coli* were identified in 63.75% (554/869) of samples; the isolation rates of *E. coli* from pet dogs, working dogs and pet cats were 53.68% (292/544), 92.72% (191/206), and 59.66% (71/119), respectively. *E. coli* was more prevalent in animals with diarrhea than in healthy animals in 2021 [70% (28/40), 54.44% (49/90), respectively].

### 2.2. Drug Resistance Phenotypes

In total, 86.46% of *E. coli* isolates exhibited resistance to at least one antibiotic. *E. coli* exhibited diverse antimicrobial resistance phenotypes, with 230 different resistance patterns detected (Appendix A). The MDR rates of *E. coli* isolates from pet dogs decreased from 76.92% from 2012–2013 to 62.42% in 2021 (*p* < 0.05), but working dogs showed the opposite trend (an increase from 41.18 to 72.86%, *p* < 0.001). There was no significant difference in the proportions of MDR *E. coli* between healthy companion animals and those with diarrhea [59.18% (29/49) vs. 60.71% (17/28), respectively, *p* > 0.05]. In pet dogs, the proportion of *E. coli* with multiple antibiotic resistance (MAR) index ≥ 0.2 gradually declined between 2012–2021, but working dogs showed the opposite trend. *E. coli* in pet cats showed more serious drug resistance than that in pet dogs in 2021 (Figure 1).

The resistance rates to different antibiotics and drug resistance patterns of isolates from different sources between 2012–2021 are shown in Appendix A and Appendix A. There was a general trend of decreasing resistance to various antibiotics in pet dog *E. coli* isolates between 2012–2021, while the opposite was true for working dog *E. coli* (Figure 2). Working dog isolates in 2021 showed higher resistance to β-lactamase inhibitors and aminoglycosides than isolates from pet dogs. Notably, almost all isolates in this study were sensitive to colistin and carbapenems, and only 1.99% (*n* = 11) were resistant to colistin and 1.99% (*n* = 11) to carbapenem antibiotics. No carbapenem resistance phenotype was found in isolates from pet cats.

### 2.3. ESBL-Producing E. coli

A total of 219 *E. coli* isolated in this study were ESBL producers, although no ESBL producer was isolated from working dogs between 2012–2013. More than half of the feline *E. coli* (60.56%) were ESBL producers. Healthy and diarrheic companion animals had a similar prevalence of ESBL producers [40.82% (20/49) vs. 39.29% (11/28), respectively, *p* > 0.05].

All ESBL-producing *E. coli* showed resistance to cefazolin, ampicillin, and piperacillin. The resistance rates of ESBL producers to fluoroquinolone antibiotics continued to decline from 2012 to 2021 (Appendix A). It is noteworthy that 2.74% (*n* = 6) of ESBL-*E. coli* in 2021 were resistant to 11–12 antimicrobial categories (Appendix A).

### 2.4. Antimicrobial Resistance Genes

All isolates with ESBL markers carried *bla*_ESBL_ genes, and no *bla*_SHV_ was detected. Our ESBL *E. coli* carried one or more *bla*_ESBL_ genes from the three enzyme families: CTX-M, TEM, and OXA-1 (Table 1), and *bla*_CTX-M_ and *bla*_CTX-M+TEM_ were dominant among the ESBL producers. *bla*_CTX-M_ was dominant between 2012–2015, but *bla*_CTX-M+TEM_ was dominant in 2021. ESBL *E. coli* carrying *bla*_CTX-M+TEM+OXA-1_ were detected between 2012–2021.

Comparisons with *bla*_CTX-M_-negative MDR *E. coli* and *bla*_CTX-M_-positive MDR *E. coli* showed significantly increased resistance to gentamicin, cephalosporins (except ceftazidime), monobactams, penicillins, amoxicillin–clavulanate, sulfonamides, and fluoroquinolones (*p* < 0.05, Table 2).

This study explored the consistency between the resistance phenotypes and resistance genes of ESBL *E. coli* for various antibiotics (Appendix A). In total, 91.18% of AG-type (resistant to amikacin and gentamicin) ESBL *E. coli* carried *rmtB*, and the rest carried *armA*; 99.45% (180/181) of tetracycline resistant strains carried *tet (A)*/*tet (B)*/*tet (C)*; *tet (A)* was detected in 85.08% (154/181) of tetracycline-resistant strains, *tet (B)* was detected in 21.55% (39/181) of tetracycline-resistant strains, and *tet (M)* was detected in 3.31% (6/181) of tetracycline-resistant strains. In addition, 8.44% (13/154) of tetracycline-resistant strains concomitantly carried *tet (A)* and *tet (B)*, and 3.90% (6/154) of tetracycline-resistant strains concomitantly carried *tet (A)* and *tet (M)*. The *tet (X1–X5)* genes were not detected in any ESBL *E. coli*. A total of 97.90% (140/143) of trimethoprim–sulfamethoxazole-resistant strains carried *sulR* genes. The *sul2* gene had the highest positivity rate, up to 79.02% (113/143), followed by *sul1* [48.95% (70/143)] and *sul3* [19.58% (28/143)], while 2.80% (4/143) concomitantly carried three *sulR* genes, 37.76% (54/143) concomitantly carried *sul1* and *sul2*, and 11.89% (17/143) concomitantly carried *sul2* and *sul3*. Some ESBL *E. coli* carrying *sulR* genes were sensitive to trimethoprim–sulfamethoxazole. Resistance genes *cmlA* or *floR* were detected in 90.68% of chloramphenicol-resistant strains, and up to 52.63% of intermediately-sensitive strains also carried *cmlA*. The carriage rate of *qnrS* in fluoroquinolone sensitive or intermediately sensitive ESBL *E. coli* (47.19%) was higher than that in resistant strains (16.15%), while *aac (6′)-Ib-cr* showed the opposite trend.

All colistin-resistant ESBL producers carried *mcr-1*, and 75% (3/4) of carbapenem resistant ESBL producers carried *bla*_NDM_, while the other strain was only resistant to imipenem. PD13dc carried both *bla*_NDM-4_ and *mcr-1*, PD82dc and PD136dc carried *bla*_NDM-5_, and all strains carrying *bla*_NDM_ were isolated from working dogs (Figure 3).

### 2.5. Phylogenetic Groups

The analysis of phylogroups showed that phylogroups A (37.77%) and B1 (30.16%) were the dominant groups among *E. coli* isolates in 2021, followed by phylogroups B2 (16.03%), F (6.79%), E (4.89%), C (2.99%), D (1.09%) and E clades (0.27%), respectively (Appendix A). The isolates from different origins exhibited different structural characteristics. Phylogroups A and B1 were the dominant groups of canine *E. coli*; however, phylogroup B2 was the dominant group in *E. coli* from companion cats, and no isolates were assigned to phylogroup D. Phylogroups A and B1 were the dominant phylogroups of canine ESBL *E. coli*, except for those isolated in 2015, and phylogroups B1 and B2 were dominant phylogroups in feline ESBL *E. coli* (Appendix A).

## 3. Discussion

Drug-resistant bacteria can be transmitted through close contact between pets and humans (such as petting, licking, or physical injuries) or through the home environment (such as contamination of food and furniture), posing a risk to children, the elderly, and pregnant or immunocompromised individuals [11]. Some high-income countries have investigated the antimicrobial resistance of *E. coli* carried by companion animals, such as the United Kingdom [12], the United States [13], Spain [14], Canada [15], Australia [16], Japan [17], and Thailand [18]. The prevalence of drug-resistant *E. coli* (86.64%) in this study was much higher than in most developed and even developing countries. Previous studies have shown that the occurrence of MDR *E. coli* in companion animals varies considerably between provinces in China, from 24 to 86% [19,20,21,22,23]. *E. coli* exhibited a variety of antimicrobial resistance patterns in our study.

Since China began to include antibiotics in the management of veterinary prescription drugs in 2013, relevant authorities have released a number of policies to curb the increase in antimicrobial resistance in animal-derived bacteria. In 2015, China issued a decision on the discontinuation of the use of four veterinary drugs (namely, lomefloxacin, pefloxacin, ofloxacin, and norfloxacin) in food animals [24]; in 2016, China banned the use of colistin sulfate to promote animals’ growth and, in 2018, China advocated avoiding the use of 3rd/4th generation cephalosporins and fluoroquinolone antibiotics in veterinary antimicrobial selection. In 2019, China began a complete ban on the addition of growth-promoting additives in feed. In our study, the resistance of *E. coli* in pet dogs to fluoroquinolones and 3rd/4th generation cephalosporins continued to decline from 2012 to 2021, the resistance to colistin decreased between 2015–2021, and the resistance of *E. coli* to various types of antibiotics in pet dogs was largely on a downward trend between 2015–2021. Although China does not allow carbapenems to be used in animals [25], a few *E. coli* with resistance to these antibiotics and related resistance genes were still detected in this experiment, which may have been caused by the transmission of drug resistance genes from humans to pets. Additionally, there have been occasional reports of companion animals carrying relevant resistance genes in recent years in China and elsewhere [25,26], so the government still needs to control these types of antibiotics strictly, because the spread of resistance to these antibiotics will cause a global health crisis [27].

It is reported that bacterial resistance is positively correlated with antibiotic use [11]. Antimicrobial resistance was generally consistent between our isolates and clinical isolates from the Chinese antimicrobial surveillance network (CHINET; https://www.chinets.com/, accessed on 21 April 2022). The possible causes are as follows: (1) The inability of companion animals to communicate the specificity and severity of their symptomatology may prompt pet owners to take precautionary and preventative measures to protect their pets, such as the use of antibiotics [28]; (2) Resistance genes may spread within the household; (3) The development of veterinary drugs is limited, and it is difficult to meet the needs of various pet diseases, so some human clinical antibiotics are used in animals; (4) Both self-medication with antibiotics and inappropriate antibiotic prescribing are prevalent in China. Self-administered antibiotics are mainly from over-the-counter purchases and leftover antibiotics from previous prescriptions [29].

As an important component of AMR *E. coli*, the prevalence of ESBL *E. coli* in companion animals varies among continents, ranging from 0.63% (Oceania) to 16.56% (Africa) in companion dogs and from 0% (Oceania) to 16.56% (Asia) in cats [30]. In this study, 41.30% (171/414) of *E. coli* isolated from companion animals were ESBL *E. coli*, which was significantly higher than the global prevalence of ESBL *E. coli* [30]. The rational use of antibiotics to reduce resistance remains a huge challenge in China. It has been reported that the prevalence of ESBL producers is higher in diarrheic animals than in healthy animals [31,32]. However, the prevalence of ESBL producers in healthy and diarrheic companion animals showed no significant difference in this study (*p* > 0.05), but the frequent and uncontrolled defecation of diarrheic pets increases the likelihood of spread of the ESBL-producing *E. coli* to the owner [33].

In 2021, ESBL *E. coli* was isolated from working dogs for the first time (34.29%). Resistance rates to most antibiotics were much lower in working dogs than in companion dogs between 2012–2013, a phenomenon that began to change in 2021. The possible reasons are as follows: (1) The use of antibiotics. Some antibiotics have been widely used in working dogs after 2020, such as penicillin and gentamicin; (2) Working dogs may acquire more drug-resistance genes from the environment during training, such as when picking up target objects from the ground. Notably, for the first time, an *E. coli* resistant to all 13 categories of antibiotics tested (except aztreonam) was isolated from working dog samples, and it carried both *mcr-1* and *bla*_NDM-4_. *E. coli* carrying both genes has been previously isolated only from broiler feces in China [34] and from human patients in north Lebanon [35]. Study has shown that the fitness cost of *bla*_NDM-4_ carriage is not lower and that its presence might indicate rare occurrence and transmission among strains of different origins [34], but relevant genes were still detected in working dogs. The other two strains of *E. coli* carrying *bla*_NDM-5_ were also isolated from working dogs, and the fitness cost of *bla*_NDM-5_ is low [34], which undoubtedly increases the difficulty of prevention and treatment of drug-resistant bacterial infections and poses a potential threat to working dogs and related practitioners. It is necessary to take action to solve the problem of the spread of drug-resistant *E. coli* in working dogs.

Many researchers have reported that ESBL *E. coli* had a higher prevalence in dogs than in cats [36,37], but the trend in this study was the opposite. This phenomenon may be caused by the following: (1) A higher usage rate of the 3rd generation cephalosporins in cats than in dogs, because cats have poor adherence to oral medication and are less likely to ingest medication in food [38]; (2) The transmission frequency of drug resistance genes between cats and owners is higher than for dogs; a study in Mexico showed that participants spent more time stroking, brushing, and hugging their cats than their dogs [39]. According to the survey data of the white paper on China’s pet industry in 2021, the number of cats surpassed that of dogs in 2021, becoming the most preferred pet; therefore, it is necessary to strengthen the monitoring of drug resistance in the intestinal flora of cats.

Most of the early ESBLs described were of the TEM or SHV type, but the CTX-M type gradually became the most prevalent ESBL type throughout the world after the emergence of the CTX-M type [40], and *bla*_CTX-M_ was the dominant *bla* gene in this study. ESBL *E. coli* resistant to amikacin in this study were resistant to gentamicin, so amikacin can be used as an indicator of aminoglycoside antibiotic resistance. Compared with *aac (6′)-Ib*, *aac (3)-II* had a higher detection rate in gentamicin-resistant strains, which is consistent with the prevalence of aminoglycoside-modifying enzymes in Chinese human clinical *E. coli* isolates [41]. The resistance of ESBL *E. coli* from dogs and cats to amikacin in this study is mainly caused by *rmtB*, which is consistent with the prevalence of 16S rRNA methylases in Asia [42]. In this study, *armA* were isolated from ESBL *E. coli* in pet cats for the first time in China, and there is only one report of *E. coli* carrying *armA* isolated from companion dogs in the world [43]. Some strains carrying *sulR* genes are sensitive to trimethoprim–sulfamethoxazole, which may be related to *dfr*, which endows bacteria with trimethoprim resistance [44]. The *sul2* genes are on large multi-resistance plasmids, indicating that it is unrealistic to eliminate the resistance genes widely distributed on mobile-genetic elements by reducing the use of antibiotics in the medium and short term [45]. The detection rates of PMQR (plasmid-mediated quinolone resistance) genes (*qnr*, *aac (6′)-Ib-cr*) in fluoroquinolone resistant strains were low. It is reported that high fluoroquinolone resistance is mainly obtained through the accumulation of PMQR genes and quinolone resistance-determining regions (QRDR) mutations [46]. PMQR genes do not confer a high level of resistance to fluoroquinolones but, rather, confer reduced susceptibility to these antimicrobial agents [44], so the prevalence of *qnrS* in sensitive and intermediate strains is much higher than that in resistant isolates. The high detection rates of *tet (A)* and *tet (B)* indicate that the main mechanism of tetracycline resistance in *E. coli* in dogs and cats in northeast China is active efflux, which is in line with the epidemic trend of tetracycline resistance genes in *E. coli* of animal origin. Resistance genes *cmlA* and *floR* are often found on plasmids and can be transferred horizontally between different species of bacteria. The high detection rate of *cmlA* in the chloramphenicol strains of intermediate sensitivity may be caused by the non-expression or low-level expression of the *cmlA* gene in vivo [47]. Most of the drug-resistant genes detected in this study exist in plasmids. The resistance rates to sulfonamides, fluoroquinolones and gentamicin were significantly higher in *bla*_CTX-M_ positive strains than in *bla*_CTX-M_ negative strains because these drug-resistant genes are on the same plasmid or can be co-transferred with *bla*_CTX-M_ [22,48,49]. Once these genes are co-transferred with *bla*_CTX-M_, they will pose a huge risk to human health.

All phylogroups were found among ESBL producers in this study, which was only previously reported in the United States and Europe [30]. It has been reported that phylogenetic group B2 was more common in diarrheic animals than in healthy animals [50], but there was no significant difference in phylogroup between diarrheic and healthy companion animals in this study (*p* > 0.05). A large number of canine *E. coli* isolated in 2021 belonged to phylogroups A and B1, which was consistent with the prevalence trend of *E. coli* phylogroups in domestic animals [51]. However, a predominance of phylogroup B2 (42.25%), followed by phylogroup B1 (35.21%) was observed in pet cats, consistent with the dominant phylogroup of *E. coli* isolated from the urine of cats in the United States [13]. B2 and D are the most common phylogenetic groups of urinary tract infection isolates in dogs and cats, and phylogroup B2 is highly associated with urinary tract infections in humans [52], so isolates in cats are even more worrying. Integrons were less frequent in B2 than non-B2 isolates [53], and there were more feline ESBL *E. coli* belonging to phylogroup B1 than phylogroup B2, suggesting a trade-off between resistance and virulence [49].

## 4. Materials and Methods

### 4.1. Sample Collection

With the approval of the Laboratory Animal Welfare and Ethics Committee of the Changchun Veterinary Research Institute, Chinese Academy of Agricultural Sciences, professionals collected 869 feces or anal swabs, placed them in a refrigerated box, and sent them with ice packs to the laboratory for testing within 24 h. The animals sampled included pet dogs (*n* = 544, collected from Animal Hospitals in Changchun, Jilin Province), pet cats (*n* = 119, collected from animal hospitals in Changchun, Jilin Province), and working dogs (*n* = 206, collected from Harbin, Heilongjiang Province). In this study, samples were collected three times. Samples from working dogs and pet dogs were collected between 2012–2013, samples from pet dogs were collected in 2015, and samples from working dogs, pet dogs, and cats were collected in 2021.

### 4.2. Isolation and Identification of E. coli

Each sample was placed in a 4 mL centrifuge tube, to which 2 mL physiological saline was added. After vortexing and mixing, 20 μL fecal suspension were placed on MacConkey agar plates (Qingdao Hope Bio-Technology Co., Ltd., Qingdao, China) and cultured at 37 °C for 16–18 h. A red single colony was selected and purified until no miscellaneous bacteria were found. Specific *E. coli* 16S rDNA primers [54] were used to identify the isolates. The biochemical indices of the PCR positive strains were further determined using the BD Phoenix-100^™^ automatic microbiological identification/drug susceptibility system (Becton, Dickinson and Company, Franklin Lakes, New York, NJ, USA). The *E. coli* strains were stored at −80 °C for further analysis (one strain per sample).

### 4.3. Antimicrobial Susceptibility and Initial ESBL Identification

The BD Phoenix-100^™^ automatic microbiological identification/drug susceptibility system was used to identify resistance phenotypes of *E. coli* isolated between 2015–2021, including aminoglycosides (amikacin and gentamicin), carbapenems (imipenem and meropenem), 1st and 2nd generation cephalosporins (cefazolin), 3rd and 4th generation cephalosporins (ceftazidime, cefotaxime, and cefepime), monobactams (aztreonam), penicillins (ampicillin and piperacillin), penicillins + β-lactamase inhibitors (amoxicillin–clavulanate and ampicillin–sulbactam), antipseudomonal penicillins + β-lactamase inhibitors (piperacillin–tazobactam), polymyxins (colistin), sulfonamides (trimethoprim–sulfamethoxazole), phenicols (chloramphenicol), fluoroquinolones (ciprofloxacin, levofloxacin, and moxifloxacin), and tetracyclines (tetracycline). In addition, for potential ESBL and carbapenemase-producing strains, the corresponding ESBL/ALERT1 resistance markers were shown in the report.

Antimicrobial susceptibility of *E. coli* isolated in 2013 was tested with the agar disk diffusion method on Mueller–Hinton (MH) agar (bioMérieux, Marcy-l’Étoile, France), according to Clinical and Laboratory Standards Institute guidelines (CLSI, 2020), including aminoglycosides [amikacin (30 μg) and gentamicin (10 μg)], carbapenems (meropenem, 10 μg), 3rd and 4th generation cephalosporins [ceftazidime (30 μg), cefotaxime (30 μg) and ceftriaxone (30 μg)], penicillins (ampicillin, 10 μg), sulfonamides (trimethoprim–sulfamethoxazole, 1.25/23.75 μg), phenicols (chloramphenicol, 30 μg), and tetracyclines (tetracycline, 30 μg). Colistin resistance was determined, according to the Danish Bacterial Resistance Surveillance System (DAN–MAP), and the MICs of *E. coli* suspected to be resistant to colistin were assessed by the broth microdilution method using Mueller–Hinton Ⅱ Broth (Solarbio, Beijing, China). *E. coli* ATCC 25922 was used as a quality control strain, and resistance (R), sensitivity (S), or intermediate (I) were attributed, according to the criteria of the CLSI. Furthermore, ESBL *E. coli* was screened by the double-disk synergy test using ceftazidime and cefotaxime with and without clavulanic acid on MH agar. When the inhibition zone diameter obtained with clavulanic acid increased by ≥5 mm, compared with that without clavulanic acid, the isolate was considered positive for ESBL.

The ESBL producers from 2012–2013 were re-tested for their antimicrobial resistance phenotype using the BD Phoenix-100^™^ automatic microbiological identification/drug susceptibility system, as described above. The MAR index was determined according to the method described by Krumperman [55].

### 4.4. Detection of ARGs (Antibiotic Resistance Genes)

All ESBL-producing *E. coli* isolated between 2012–2021 were screened for ARGs, including (1) extended-spectrum β-lactamase genes *bla*_CTX-M_, *bla*_SHV_, *bla*_OXA-1_, and *bla*_TEM_, (2) tetracycline resistance genes *tet (A)*, *tet (B)*, *tet (M)*, and *tet (X1–X5)*, (3) sulfonamide resistance (*sulR*) genes *sul1*, *sul2*, and *sul3*, (4) chloramphenicol resistance genes *cmlA* and *floR*, (5) aminoglycoside resistance genes *aac (3)-II*, *aac (6′)-Ib*, *armA*, and *rmtB*, (6) fluoroquinolone resistance genes *qnrA*, *qnrB*, *qnrC*, *qnrS*, and *aac (6′)-Ib-cr*. Carbapenemase and colistin resistance genes (*bla*_NDM_, *bla*_KPC_, *bla*_IMP_, and *mcr-1–5*) were screened depending on the antimicrobial resistance (Appendix A). The amplicons obtained were evaluated and analyzed using a UV gel imaging system after electrophoresis. The positive amplicons were randomly selected for sequencing (Comate Bioscience Company Ltd., Changchun, China), and the sequences were analyzed using the BLAST program available at the National Center for Biotechnology Information (http://www.ncbi.nlm.nih.gov/BLAST/, accessed on 12 June 2022).

### 4.5. Phylogenetic Grouping

DNA templates were made by the boiling method [56]. All ESBL producers between 2012–2021 and *E. coli* isolates obtained in 2021 were amplified with *chuA*, *yjaA*, T*spE4C2*, and *arpA*, according to the multiplex PCR method described by Clermont et al. [57]. The phylogroups (A, B1, B2, C, D, E, and F) were identified according to the different amplified fragment sizes. Sterile distilled water was the negative control, and the positive control was the strain template mixture preserved in this trial.

### 4.6. Statistical Analysis

We tested for statistical significance using the chi-square test of independence in SPSS 20.0 software (IBM, Armonk, New York, USA). Pearson’s chi-squared test, Yate’s correction for continuity, and Fisher’s exact test were used in statistical significance when no/one/two or more cell counts in a 2 × 2 table were less than 5, respectively. A *p*-value < 0.05 was regarded as the criterion for statistical significance.

## 5. Conclusions

With the restrictions on the use of antibiotics in China, the antimicrobial resistance of *E. coli* in companion dogs is on the decline. Companion cats may be potential hosts for *E. coli*, causing extraintestinal disease in humans. Compared with pet dogs, the problem of drug-resistant bacteria carried by pet cats has become prominent; in particular, the high prevalence of ESBL *E. coli*, which belong to the B2 group in fecal samples from pet cats deserves further study. Our data helped to fill the gap in the knowledge of intestinal *E. coli* resistance in working dogs. All ESBL *E. coli* carrying *bla*_NDM_ were isolated from working dogs, and one of them carried *mcr-1* and *bla*_NDM-4_. The problem of drug-resistant bacteria in working dogs is often neglected and needs to be paid attention. As animals in close contact with humans, the public health risk associated with dogs and cats carrying MDR zoonotic pathogens requires continued concern.

## Figures and Tables

**Figure 1 antibiotics-11-01506-f001:**
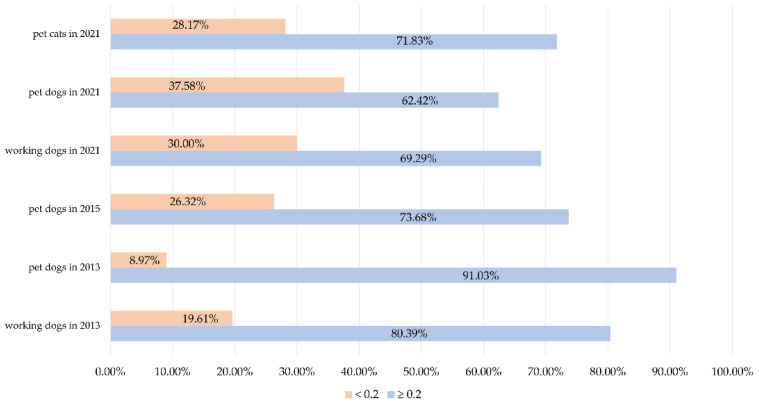
MAR index of *E. coli* of different origins and years.

**Figure 2 antibiotics-11-01506-f002:**
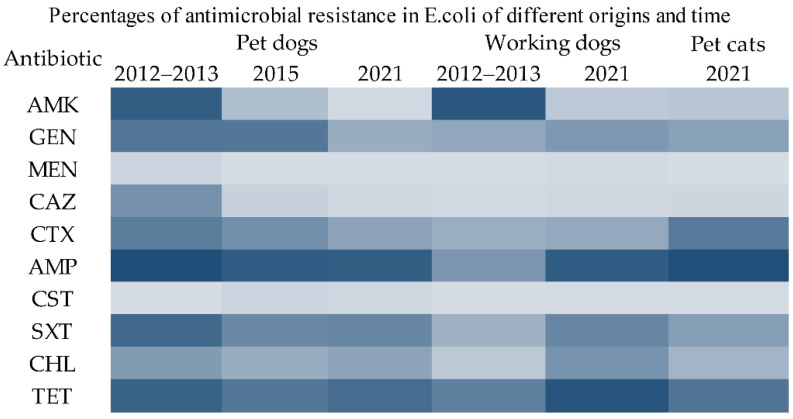
Trends of resistance to various antibiotics of 554 *E. coli* isolates from different origins between 2012–2021 (the depth of color of the rectangle represents the degree of resistance of the isolates to the antibiotic, and the darker the color, the stronger the drug resistance). AMK, amikacin; GEN, gentamicin; MEN, meropenem; CAZ, ceftazidime; AMP, ampicillin; CST, colistin; SXT, trimethoprim-sulfamethoxazole; CHL, chloramphenicol; TET, tetracycline. Notes: the antibiotics listed in Figure 2 only include the antibiotics tested against all *E. coli*.

**Figure 3 antibiotics-11-01506-f003:**
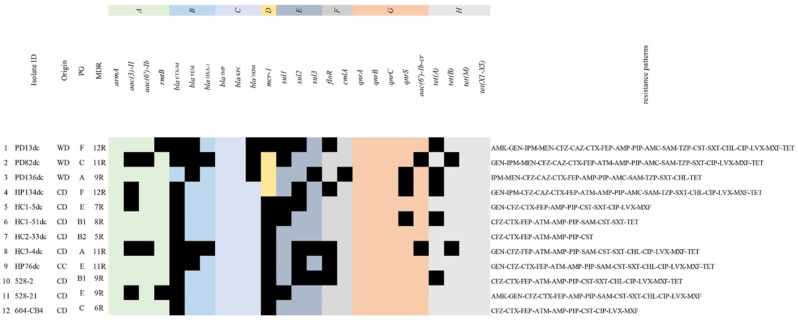
Resistance gene profiles and resistance patterns of 12 ESBL *E. coli*, which were resistant to carbapenems or colistin. Black squares indicate gene presence. A. Aminoglycoside resistance genes; B. *bla*_ESBL_; C. carbapenem resistance genes; D. colistin resistance genes; E. sulfonamide resistance genes; F. chloramphenicol resistance genes; G. fluoroquinolone resistance genes; H. tetracycline resistance genes; WD. working dog; CD. companion dog; CC. companion cat; PG. phylogenetic groups.

**Table 1 antibiotics-11-01506-t001:** *bla*_ESBL_ types of ESBL producers from 2012 to 2021.

*bla* _ESBL_	2012–2013	2015	2021	2021	2021
	Pet Dogs	Pet Dogs	Pet Dogs	Pet Cats	Working Dogs
*bla* _CTX-M_	20	15	29	18	17
*bla* _TEM_	0	0	0	0	2
*bla* _CTX-M+TEM_	13	10	30	20	28
*bla* _CTX-M+OXA-1_	4	1	0	4	0
*bla* _CTX-M+TEM+OXA-1_	4	1	1	1	1

**Table 2 antibiotics-11-01506-t002:** Prevalence of the *bla*_CTX-M_ gene and antibiotic resistance of MDR *E. coli* in 2021.

Antibiotic	*bla*_CTX-M_ Positive (*n* = 149)	*bla*_CTX-M_ Negative (*n* = 102)	*p*-Value
Amikacin	13.42%	9.80%	0.385
Gentamicin	61.74%	32.35%	<0.001 **
Imipenem	2.01%	3.92%	0.447
Meropenem	1.34%	0.98%	1.000
Cefazolin	100.00%	12.75%	<0.001 **
Ceftazidime	7.38%	2.94%	0.132
Cefotaxime	91.28%	0.98%	<0.001 **
Cefepime	86.58%	0.98%	<0.001 **
Aztreonam	60.40%	0.00%	<0.001 **
Ampicillin	100.00%	95.10%	0.010 *
Piperacillin	100.00%	81.37%	<0.001 **
Amoxicillin–Clavulanate	4.03%	10.78%	0.036 *
Ampicillin–Sulbactam	41.61%	33.33%	0.185
Piperacillin–Tazobactam	3.36%	0.98%	0.406
Colistin	4.03%	1.96%	0.479
Trimethoprim–Sulfamethoxazole	60.40%	72.55%	0.047 *
Chloramphenicol	53.69%	47.06%	0.302
Ciprofloxacin	47.65%	32.35%	0.016 *
Levofloxacin	46.31%	30.39%	0.011 *
Moxifloxacin	52.35%	33.33%	0.003 *
Tetracycline	85.91%	91.18%	0.206

* *p* < 0.05, ** *p* < 0.001.

## Data Availability

The data that support the findings of this study are available from the corresponding author upon reasonable request.

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
