# Peer review of "Antimicrobial Resistance and Prevalence of Extended Spectrum β-Lactamase-Producing Escherichia coli from Dogs and Cats in Northeastern China from 2012 to 2021"

_antibiotics, 2022, doi:10.3390/antibiotics11111506_

Round 1
Reviewer 1 Report
The authors did an excellent job of providing information on the prevalence of antimicrobial-resistant Escherichia coli in dogs and cats in North-eastern China from 2012 to 2021. The presence of resistant bacteria, especially ESBL-producing E. coli in dogs and cats poses a serious health threat, by being transferred to humans easily. I believe that the findings from this research will help in taking initiatives for the selection of antibiotics in animals, especially in dogs and cats.
However, I have a few comments, which should be addressed before publication.
Comments
Abstract
Page 1
Line 15: Please remove E. coli from the bracket. You don’t have to use it. You can normally write “E. coli” after the first use of “Escherichia coli”.
Line 18: “and antibiotic susceptibility profiles…..”. Here add “their” after and.
Line 19: Please give the full form of ESBL here.
Line 19: Please avoid numerals at the start of a sentence. So, you should either write “Five hundred fifty-four” or rephrase the sentence.
Line 20: Please omit “fecal” from here. Just write “…869 samples..”
Line 21: Please write “Resistance genes” before “blaCTX-M”
Line 29: All the keywords are available in the title of the manuscript. Please use a few different keywords which are relevant to your research. Also, you can use “AMR” instead of antimicrobial resistance, “E. coli” instead of Escherichia coli, and “ESBL” instead of extended-spectrum β-lactamase.
Introduction
Page 2
Line 3: Please write Escherichia coli instead of E. coli here. The scientific name of an organism should be full-formed at the start of a paragraph.
Line 4: Please mention the short form of multidrug-resistant here, because you have used MDR later.
Line 5-6: Not more than three, it should be “more than two” or “at least three” or “three or more”. Please correct it.
Line 8: Please give the full form of ESBL here.
Results
In your results section, you’ve only mentioned 2012-13, 2015, and 2021. What’s about 2014 and 2016-2020? Didn’t you find any E. coli isolates during those years? If yes, please mention it in the results section. Or if you didn’t take data from those years, please mention the purposes in the materials and methods.
Materials and Methods
Page 10
Line 40: You used the Chi-square test to compare the differences. What kind of Chi-square test did you use here? Chi-square goodness of fit test or Chi-square test of independence? Please mention it here. Also, I think you had to do Fisher’s exact test for showing variations where the number of counts was less than 5 (especially, during performing analysis to show variations between blaCTX-M-positive isolates and blaCTX-M-negative isolates). When one or more cell counts in a 2×2 table are less than 5, Fisher’s exact test is typically used instead of the Chi-square test of independence. If you did it, please mention it here.
Reviewer 2 Report
This study explored the impact of changes in antibiotic use trends in recent years on E. coli drug resistance phenotypes by investigating drug resistance of E. coli in dogs in Northeast China from 2012 to 2021, and also investigated the prevalence of drug resistance phenotypes and drug resistance genes of ESBL-E. coli in dogs and cats, as well as enriching the drug resistance data related to E. coli in working dogs. The manuscript is well written, however, there are a few comments that need to be addressed to improve the manuscript:
- Please add a table in the samples collection section to distribute your samples.
- In section 4.3, you need to add the concentration of each antibiotic.
- Discussion is very long. please try to shorten it.
- Provide the correlation between phenotypic and genotypic analysis.
- It's better to change figure 2 to a table.
- Provide the resistance pattern as a trend graph for each year.
-
Reviewer 3 Report
Antibiotics-1923183
Antimicrobial Resistance and Prevalence of Extended Spectrum β-lactamase-producing Escherichia coli from Dogs and Cats in Northeastern China from 2012 to 2021
Major comments:
Although the study is interesting, the intent of the research, methodology and the results are not present clearly. Were these samplings done on yearly basis from 2012-2021 with the purpose of understanding trends in antimicrobial susceptibilities of E. coli from the pets and working animals?
It appears from the methods section that the pet dog samples and working dog samples were collected from two different provinces. What is the rationale behind this?
Further, the data on pet cats is mostly restricted to 2021. For example, you have compared pet cats and dog data of 2021 in Figure 1 and Table 1. What about the previous years?. As I could understand from the table 1, you have presented data of 2012-2013 for pet dogs, and of 2021 for pet dogs, working dogs and pet cats. The methodology sections also does not describe the sampling period. In discussion too, it is important to discuss the trend from 2021 to 2021. However, it is more focussed on describing the situation in 2021, from antibiotic susceptibility profiles to the phylogroups.
Similarly, table 2 summarises the susceptibility data of only 2021. Again, it is not clear what you are testing by the test of statistical significance. For example, you have listed blaCTX positive and negative percentages against non beta-lactam antibiotics? How is this relevant? Throughout the manuscript, the data is presented in an assorted manner. To cite another example, Figure 3 shows antibiotic resistance and resistance gene profiles of 12 ESBL-positive resistant to carbapenem and colitin E. coli. Did you have only 12? .
Page 8, L2: this statement is in contrast with the results in page 2, L36. ESBL is higher, but MDR is lower between health and diarrheic animals?
Page 8, 4-6: How? Because these bacteria are protected from sunlight?
With reference to Figure 1, how would there be more resistance to 6-9 antimicrobials compared to 3-5 antimicrobials. These 6-9 antibiotics do not include those which are in 3-5?. I think the best way to represent increase in antimicrobial resistance is through MAR index.
Please note, disk diffusion test is insufficient to determine colistin susceptibility.
General comments
Line 27: “easlier”?
L39: “rarely cause clinical disease”. E. coli infections are not rare. You might consider rephrasing this sentence.
L39: How many E. coli isolates were recovered from pet cats?
P9, L31: Please rephrase this sentence. Are there standard procedures to isolate E. coli from pets? Did you mean, you streaked 20 µl on MacConeky agar?. On MacConkey, you find red colonies, not purplish red colonies.
P2, L29: There is no data on how many were healthy or diarrheic animals.
P2, L33-34: Decreased or increased with respect to what?
Fig 1 shows companion dogs. Please use one terminology; either pet dogs or companion dogs
Generally line diagrams are employed to represent changes with respect to the same parameter over a period of time or similar data. I suggest the authors to show these differences with respect to various antibiotics in a different way (e.g heat map). This figure is misleading as it shows variations in susceptibilities to different antibiotics by different groups. Line diagram will be more appropriate to show changes in ESBL production over the study period in different groups.
Round 2
Reviewer 2 Report
The authors addressed my comments, however, they still need to present the resistance pattern in a trend graph. the manuscript needs English proof
Reviewer 3 Report
The authors have made extensive changes as suggested in my previous review. However, I could not find a cleaner version of the mansucript (without track changes). However, the corrected senetences, a couple of which are pointed out below, seem to be grammatically incorrect.
L20: "There are 554 E. coli isolates were detected in 869 samples "
L27-28: "Companion cats may be more susceptible colonized with ESBL-E. coli.
Section 2.2: Why the percentage resistance values have changed? There appears to be chnages in the number of resistant isolates.
Figure 1. Authors might consider using a colored illustration
Table1. Please align the years 2015 and 2021 properly. It is not clear, which column corresponds to which year.
P6, line 11: Italicize sulR
P8, L32: "carried" . In the same line, please replace "which carries" with "carrying"
P11, L33: Please remove an extra ")"
